# Multi-Dimensionality Immunophenotyping Analyses of MAIT Cells Expressing Th1/Th17 Cytokines and Cytotoxic Markers in Latent Tuberculosis Diabetes Comorbidity

**DOI:** 10.3390/pathogens11010087

**Published:** 2022-01-12

**Authors:** Gokul Raj Kathamuthu, Nathella Pavan Kumar, Kadar Moideen, Chandrakumar Dolla, Paul Kumaran, Subash Babu

**Affiliations:** 1National Institutes of Health-NIRT-International Center for Excellence in Research, Chennai 600031, India; pavankumarn@nirt.res.in (N.P.K.); sbabu@niaid.nih.gov (S.B.); 2National Institute for Research in Tuberculosis (NIRT), Chennai 600031, India; kadar.m@nirt.res.in (K.M.); ckdolla@gmail.com (C.D.); ppaulkumaran@nirt.res.in (P.K.); 3Laboratory of Parasitic Diseases, National Institute of Allergy and Infectious Diseases, National Institutes of Health, Bethesda, MD 20892-0425, USA

**Keywords:** Th1 cells, Th17 cells, cytotoxic markers, UMAP analysis, flow cytometry

## Abstract

Mucosal-associated invariant T (MAIT) cells are innate like, and play a major role in restricting disease caused by *Mycobacterium tuberculosis* (Mtb) disease before the activation of antigen-specific T cells. Additionally, the potential link and synergistic function between diabetes mellitus (DM) and tuberculosis (TB) has been recognized for a long time. However, the role of MAIT cells in latent TB (LTB) DM or pre-DM (PDM) and non-DM (NDM) comorbidities is not known. Hence, we examined the frequencies (represented as geometric means, GM) of unstimulated (UNS), mycobacterial (purified protein derivative (PPD) and whole-cell lysate (WCL)), and positive control (phorbol myristate acetate (P)/ionomycin (I)) antigen stimulated MAIT cells expressing Th1 (IFNγ, TNFα, and IL-2), Th17 (IL-17A, IL-17F, and IL-22), and cytotoxic (perforin (PFN), granzyme (GZE B), and granulysin (GNLSN)) markers in LTB comorbidities by uniform manifold approximation (UMAP) and flow cytometry. We also performed a correlation analysis of Th1/Th17 cytokines and cytotoxic markers with HbA1c, TST, and BMI, and diverse hematological and biochemical parameters. The UMAP analysis demonstrated that the percentage of MAIT cells was higher; T helper (Th)1 cytokine and cytotoxic (PFN) markers expressions were different in LTB-DM and PDM individuals in comparison to the LTB-NDM group on UMAP. Similarly, no significant difference was observed in the geometric means (GM) of MAIT cells expressing Th1, Th17, and cytotoxic markers between the study population under UNS conditions. In mycobacterial antigen stimulation, the GM of Th1 (IFNγ (PPD and WCL), TNFα (PPD and WCL), and IL-2 (PPD)), and Th17 (IL-17A, IL-17F, and IL-22 (PPD and/or WCL)) cytokines were significantly elevated and cytotoxic markers (PFN, GZE B, and GNLSN (PPD and WCL)) were significantly reduced in the LTB-DM and/or PDM group compared to the LTB-NDM group. Some of the Th1/Th17 cytokines and cytotoxic markers were significantly correlated with the parameters analyzed. Overall, we found that different Th1 cytokines and cytotoxic marker population clusters and increased Th1 and Th17 (IL-17A, IL-22) cytokines and diminished cytotoxic markers expressing MAIT cells are associated with LTB-PDM and DM comorbidities.

## 1. Introduction

Latent tuberculosis (LTB) does not have any of the clinical symptoms of active tuberculosis (ATB): it is defined by the presence of an immunological reaction to *Mycobacterium tuberculosis* (Mtb) antigens as determined by a tuberculin skin test (TST) or an interferon-gamma release assay (IGRA) [1,2,3,4]. However, only a small fraction (5–15%) of LTB-infected individuals have a lifetime risk of reactivation of ATB, with the majority of progression occurring within the first 2–5 years following initial exposure [5,6]. Therefore, LTB serves as a key reservoir for the development of ATB [7]. In addition, both diabetes mellitus (DM) and human immunodeficiency virus (HIV) co-morbidities are most strongly attributed to enhanced risk of developing ATB [8,9]. Likewise, the incidence of pre-diabetes (PDM) is increasing, and by the year 2030, around 470 million individuals will develop PDM; depending on the demographic and geographic situation, 5–10% of them will develop overt diabetes each year [10,11,12]. PDM was correlated with increased risk of LTB infection [7] and is known to be connected with pulmonary TB (PTB) patients with respiratory ailments [8] and dysregulates cytokine immune responses [9]. Furthermore, the incidence of PDM is 25% higher in PTB individuals [13]. Therefore, in addition to DM, PDM plays a crucial function in the interconnection among metabolic disorders and PTB and LTB disease. 

Mucosal-associated invariant T (MAIT) cells are unconventional innate T cells and, in the circulation, they represent 1–10% of the T-cell population and are regulated by MHC-related molecule 1 (MR1) [11]. MAIT cells exhibit antimicrobial activity by directly killing the bacteria both extracellular and intracellularly, and indirectly by neutrophil recruitment, heightened bactericidal activity of phagocytes, IFNγ production from dendritic cells (DC), and monocyte-to-DC differentiation [14,15,16,17]. MAIT cells are vastly present in the alveoli of the lungs and bronchoalveolar lavage (BAL) fluid, as well as express CD69 activation marker after Mtb infection [18,19]. Both BCG-vaccinated and Mtb-infected nonhuman primates were able to induce proliferation of MAIT cells [19]. Several other studies have also revealed that MAIT cell frequencies are significantly altered during active TB disease [20,21,22]. Active TB disease is characterized by higher MAIT cell expression of PD-1 molecule, which is reversed following anti-TB treatment [23]. Increased MAIT cells were more strongly associated with LTB individuals than healthy controls [24]. It was also reported that the MAIT cell number was inversely correlated with systemic inflammatory and positivity of TB disease markers in the sputum [21]. Hence, MAIT cells potentially mediate the correlates of protective immunity during PTB disease and, therefore, either exhausted or diminished MAIT cell abundance may lead to reactivation of Mtb infection. Nevertheless, the function of MAIT cells in LTB and DM, PDM, and NDM comorbidities is not known. Hence, our study aim was to examine the role of Th1/Th17 cytokines and cytotoxic markers expressing MAIT cells in LTB disease with PDM, DM, and NDM comorbidities to understand whether DM or PDM comorbidities have a significant impact on the modulation of those immune markers. Thus, we found that LTB-PDM and LTB-DM comorbidities are characterized by differential expression of Th1 cytokines, cytotoxic (PFN) immune clusters, and altered Th1/Th17 (IL-17A and IL-22) cytokines and cytotoxic markers expressing MAIT cells upon Mtb antigen stimulation compared to unstimulated and positive control antigen stimulation.

## 2. Methods

### 2.1. Study Ethics

The study was sanctioned by the National Institute of Research in Tuberculosis (NIRT) Internal Ethics Committee (IEC2011013). We obtained informed and written consent from the recruited study participants. 

### 2.2. Study Participants

The demographic features of the study population are shown in Table 1. In total, a group of 60 individuals were recruited with LTB infection and they were separated into 3 groups (20 individuals in each group) as follows: diabetes mellitus (DM), pre-DM (PDM), and non-DM (NDM). LTB positivity was confirmed by both tuberculin skin test (TST) and a QuantiFERON TB-Gold in-tube ELISA with nonappearance of abnormalities in chest radiography or devoid of any pulmonary symptoms. Positive TST was defined as an induration at the site of tuberculin inoculation of at least 5 mm in diameter. Similarly, type 2 DM (glycated hemoglobin (HbA1c) levels > 6.5%, random blood glucose > 200 mg/dL), PDM (HbA1c levels >5.7% to <6.4% and random blood glucose 140–199 mg/dL), and NDM (HbA1c levels < 5.7% random blood glucose < 140 mg/dL) was diagnosed on the basis of glycated hemoglobin (HbA1c) levels and random blood glucose according to the American Diabetes Association criteria. All study individuals were HIV-negative, BCG-vaccinated, and not under any steroid usage or did not exhibit any signs or symptoms of related lung or systemic disease.

### 2.3. Isolation and Thawing of Peripheral Blood Mononuclear Cells (PBMCs)

Peripheral blood samples were collected in sodium heparin tubes and PBMCs were isolated by a Ficoll-Paque (GE Life Sciences) density gradient. The cells were centrifuged for 30 min at 400× *g* at 4 °C. The resulting PBMC layer was decanted, washed twice, counted, and stored at –80 °C using fetal bovine serum (FBS) and dimethyl sulfoxide (DMSO). The cells were thawed in 37 °C water bath using thawing (9 parts Roswell Park Memorial Institute (RPMI) 1640 + 1-part FBS) medium. After thawing, the cells were rested for 2 h at 37 °C in 5% CO_2_, and then the cell viability was assessed using a trypan blue exclusion assay. Overall, we observed 80–85% cell confluency in our study cohort.

### 2.4. Antigens and Stimulation

The following mycobacterial antigens were used in the study: purified protein derivative (PPD, 10 μg/mL; Statens Serum Institute), whole-cell lysate (WCL, 1 μg/mL), and phorbol 12-myristate 13-acetate (P) and ionomycin (P/I, 12.5 and 125 ng/mL, respectively; Calbiochem) was used as the positive control stimulant. The PBMC culture experiments were carried out to measure the frequencies of MAIT cells expressing cytokines and cytotoxic markers. Briefly, the PBMCs were mixed with complete RPMI medium (CRPMI, RPMI 1640 provided with penicillin–streptomycin (100 U/100 mg/mL), L-glutamine (2 mM), and HEPES (10 mM) (Invitrogen, San Diego, CA, USA) buffer). The cells were transferred into cell culture plates (12-well culture plates, Costar, Corning) and stimulated with the respective antigens and incubated for 18 h at 37 °C in 5% CO_2_. After 2 h post-incubation, Fast Immune^TM^ brefeldin A solution (10 μg/mL, BD Biosciences) was added to the culture plates. After incubation, cells were precisely transferred into 15 mL sterile falcon tubes and centrifuged for 10 min at 400 relative centrifugal force (RCF) or G force. The cells were rinsed in 1× PBS, dissolved in Permeabilization Buffer^TM^, and utilized for immunophenotyping after the supernatants were collected. 

### 2.5. PBMC Staining and Flow Cytometry

PBMCs were first stained with surface antibodies and then incubated at 4 °C in dark conditions for about 30–60 min. Then, the cells were washed with BD permeabilization/wash buffer followed by staining with intracellular antibodies, and they were incubated for 2 h at 4 °C. After incubation, cells were washed once again with permeabilization buffer and acquisition was performed. The antibodies were purchased from BD Pharmingen, Invitrogen (Biosciences), Miltenyi Biotech, and R&D systems: CD3 (SK7 and AmCyan), CD4 (RTA-T4 and APC-H7), CD8 (PECy7), TNFα (FITC), IFNγ (B27 and PE), IL-2 (MQ1-17H12 and APC), IL-17A (N49-653 and FITC), IL-22 (22URPI and PE), IL-17F (O33-782, Alexafluor 647), perforin (FITC), granzyme B (GB11, Alexafluor 647), granulysin (DH2 and PE), CD161 (HP-3G10 and PB), and TCR Vα7.2 (PerCp-Vio700). Furthermore, the cells were incubated for 2 h at 4 °C. Then, the cells were washed with 2 mL of Permeabilization Buffer^TM^ and dissolved with 1× PBS. The cells were acquired using eight-color cytometry on a FACSCanto II flow cytometer with FACSDiva software v.6 (Becton Dickinson and Company, Cockeysville, MD, USA). Forward vs. side scatter was used to set the lymphocyte gating, and 50,000 gated lymphocyte events were acquired. The gating strategy for MAIT cells expressing cytokines, and cytotoxic and activation markers was determined by FMO, and the frequency data of cytokines and cytotoxic markers are presented as geometric means (GMs). 

### 2.6. Data Analysis

Flowjo 3 (TreeStar Inc., Ashland, OR, USA) software was used to measure the frequencies (presented as GM) of different cellular subsets. Uniform manifold approximation and projection (UMAP) analysis was performed using FlowJo plugins by downsampling the 5000-cell CD3^+^ population gated from lymphocytes. We gated the single cells through FSC A versus FSC H gating prior to gating the positive MAIT cells. From the CD3^+^ population, we gated the MAIT cells using TCR Vα7.2^+^ and CD161^+^ only (due to being the subject of interest) and not the other Vα7.2^−^/CD161^+^, Vα7.2^+^/CD161^−^, and Vα7.2^−^/CD161^−^ population, and report them as immune clusters. All the other statistical analyses were performed using GraphPad PRISM (version 8) software (GraphPad Software, Inc., San Diego, CA, USA). The geometric means (GMs) were used to assess the central tendency, and intergroup comparisons were measured by the nonparametric Kruskal–Wallis test with Dunn’s multiple comparison test. Correlation analysis (with *p* < 0.05 considered significant) was performed using nonparametric Spearman correlation using GraphPad PRISM (version 9) software. Principal component analysis (PCA) was performed using JMP14 software. 

## 3. Results

### 3.1. UMAP Analysis of MAIT Cell Expressing Th1 Cytokines in LTB Comorbidities

The multi-dimensionality representative plots of gated MAIT cells expressing Th1, Th17, and cytotoxic markers for LTB (DM, PDM, and NDM) comorbidities are depicted by the unsupervised clustering of flow data using two-dimensional UMAP (UMAP_1 vs. UMAP_2) analysis. The percentage of MAIT cells (Th1 and Th17 cytokines, and cytotoxic markers) was lower in the LTB-NDM group than in the LTB-PDM and LTB-DM individuals (Figure 1). We display the ungated and gated expression (Appendix A Appendix A) of MAIT cells from CD3^+^ lymphocytes (Appendix A–C) and Th1 (IFNγ, IL-2, and TNFα) immune clusters in terms of percentages and frequencies (Appendix A–F) along with their distribution pattern (Appendix A–H) are presented (Appendix A). We also provide the UMAP statistical values of MAIT cells expressing Th1, Th17 cytokines, and cytotoxic markers in Appendix A Appendix A. The expression of Th1 (IFNγ, IL-2, and TNFα) immune clusters in MAIT cells differed upon Mtb antigen (both PPD and WCL) stimulation in the LTB-PDM group in comparison to the LTB-DM and LTB-NDM groups on UMAP analysis. In addition, these differences in the Th1 immune clusters were not observed in either the unstimulated (UNS) group or the positive control (P/I) with antigen stimulation (Figure 2A–C).

### 3.2. Altered GM of MAIT Cells Expressing Th1 Cytokines 

We show the MAIT (Th1, Th17, cytotoxic marker) cell gating strategy and representative plots in Appendix A Appendix A. The percentages of the CD3^+^ populations between UNS and P/I antigen stimulation are shown in Appendix A Appendix A. We examined the MAIT cells expressing Th1 cytokines in the LTB group with DM, PDM, and NDM comorbidities. We performed multicolor flow cytometry to define the frequencies (represented as the geometric mean (GM)) of unstimulated (UNS) and mycobacterial-antigens-stimulated MAIT cells expressing Th1 (IFNγ, IL-2, and TNFα) cytokines (Figure 3). In the unstimulated (UNS) condition, the GM of MAIT cells expressing Th1 (IFNγ, IL-2, and TNFα) cytokines was not significantly altered in LTB-DM and LTB-PDM individuals compared LTB-NDM individuals. In contrast, LTB-DM individuals were associated with increased GM of MAIT cells expressing IFNγ (PPD and WCL) and decreased IL-2 (PPD) cytokines in comparison with LTB-PDM and/or LTB-NDM individuals. LTB-PDM individuals were associated with an increased GM of TNFα (PPD)-expressing MAIT cells in comparison to LTB-NDM individuals. Finally, the GM of MAIT cells expressing Th1 cytokines upon PI stimulation was not significantly different between LTB coinfected individuals (Figure 3). 

### 3.3. UMAP Analysis of MAIT Cell Expressing Th17 Cytokines in LTB Comorbidities

We show the ungated and gated expressions of MAIT cells from CD3^+^ lymphocytes and Th17 (IL-17A, IL-17F, and IL-22) immune clusters in percentages and frequencies along with their distribution pattern in Appendix A Appendix A. The expressions of Th17 (IL-17A, IL-17F, and IL-22) immune clusters without stimulation or with the Mtb antigen (both PPD and WCL) stimulation and positive control (P/I) antigen stimulation did not significantly different on dimensionality analysis (Figure 4A–C).

### 3.4. MAIT Cell Expressing Th17 Cytokines Were Not Significantly Different among LTB Comorbidities

We show the GM of UNS, mycobacterial-antigen-specific (PPD and WCL) and positive antigen-stimulated MAIT cell expressing Type 17 cytokines in LTB individuals with DM, PDM, and NDM comorbidities in Figure 5. The GMs of MAIT cells expressing Th17 (IL-17A, IL-17F, and IL-22) cytokines were not significantly different in the LTB-DM and PDM individuals compared to the LTB-NDM-affected individuals under UNS conditions (Figure 5). In contrast, LTB-DM and/or PDM individuals were associated with an elevated GM of MAIT cells expressing IL-17A (PPD and WCL), IL-17F (PPD), and IL-22 (in WCL) cytokines compared to LTB-NDM individuals. Finally, the GM of MAIT cells expressing Th17 cytokines upon P/I stimulation did not significantly differ between the study individuals (Figure 5). Thus, LTB-coinfected individuals were associated with a higher GM of IL-17 family cytokines expressing MAIT cells.

### 3.5. UMAP Analysis of MAIT Cell Expressing Cytotoxic Markers in LTB Comorbidities

Appendix A Appendix A displays the ungated and gated expression of MAIT cells from CD3^+^ lymphocytes (Appendix A) and cytotoxic marker (PFN, GZE B, and GNLYSN) immune clusters, expressed in percentages and frequencies (Appendix A) along with their distribution pattern (Appendix A) (Appendix A Appendix A). The cytotoxic markers’ expression of MAIT cells differed upon Mtb antigen (both PPD and WCL) stimulation in the LTB-PDM and LTB-DM group compared to the LTB-NDM individuals upon UMAP analysis; no significant difference was observed among the study individuals in the unstimulated (UNS) and positive control (P/I, except GZE B) antigen-stimulation groups (Figure 6A–C).

### 3.6. MAIT Cells Expressing Cytotoxic Markers Were Significantly Reduced in LTB-PDM Individuals

We defined the GM of UNS, mycobacterial antigen-specific (PPD and WCL) and positive antigen-stimulated MAIT cells expressing cytotoxic markers (perforin (PFN), granzyme B (GZE B), and granulysin [GNLSN]) in LTB individuals coinfected with DM, PDM, and NDM comorbidities. As shown in Figure 7, in the unstimulated condition, the GM of MAIT cells expressing cytotoxic (GZE B and GNLSN) markers was not significantly different in LTB individuals with DM, PDM, and NDM comorbidities. In contrast, the GM of MAIT cells expressing cytotoxic markers (PFN, GZE B, and GNLSN) was significantly lower upon stimulation with PPD and WCL antigens in individuals with LTB-DM comorbidity compared to LTB-PDM and LTB-NDM comorbidities. With P/I stimulation, the GM of cytotoxic (PFN, GZE B, and GNLSN) markers was not significantly different between LTB-coinfected individuals (Figure 7). Thus, the LTB-DM comorbidity group was associated with a diminished GM for the PFN cytotoxic marker.

### 3.7. Correlation and PCA Analysis of MAIT Cells Expressing Th1, Th17 Cytokines and Cytotoxic Markers in LTB Coinfected Individuals

We performed a correlation analysis of MAIT cells expressing Th1, Th17 cytokines, and cytotoxic marker frequencies with HbA1c, TST, body mass index (BMI), hematological (red blood cells (RBCs), white blood cells (WBCs), hemoglobin (Hb), lymphocytes, neutrophils, monocytes, eosinophils, and basophils), and biochemical (urea and creatinine) parameters (Figure 8A I–III). Upon UNS, Mtb antigen, and P/I stimulation, we found a significant positive or negative correlation (*r* values included in the box) for some of the Th1, Th17 cytokines, and cytotoxic markers with different parameters analyzed; their *p*-values (significance is highlighted in red) are given in Appendix A. In PCA, we did not observe any major discrimination between the study populations with the parameters analyzed (Figure 8B IV–VI).

## 4. Discussion

MAIT cells are the effective responders to mycobacterial infection; a previous study described their relationship after mycobacterial exposure in macaques [19]. Similarly, the relationship between DM and the associated risk of pulmonary TB has been widely explored [25]. Previous data from our group showed that LTB-PDM individuals have decreased frequencies of monofunctional CD4^+^ and CD8^+^ cells expressing Th1/Tc1, Th2/Tc2, and Th17/Tc17 cytokines at UNS and/or following Mtb antigen stimulation compared to LTB-NDM individuals [26]. LTB individuals are also characterized by decreased CD4^+^ expressing Th1, Th2, and Th17 cytokines, potentially mediated by both IL-10 and TGFβ [27]. LTB-DM is also associated with reduced CD8^+^ T-cell cytokines upon induction with Mtb antigens [28]. Type 1, Type 17, and certain proinflammatory cytokines are significantly reduced in LTB disease [29]. More importantly, PDM individuals are associated with a heightened risk of LTB and active TB disease [30,31].

T-cell-mediated immune responses, and particularly CD4^+^ T cells expressing Th1 cytokines, are the important mediator of an efficient immune response against TB and clinical outcomes [32,33]. Moreover, multifunctional CD4^+^ T cells, which are correlated to the defensive immune response, were shown to be higher in LTB infection than active TB disease and were inversely correlated with reduced bacterial load [34,35]. Similarly, CD4^+^/Vα7.2^+^/CD161^++^ MAIT cells were significantly enhanced in LTB individuals compared to active TB cases [36]. A previous study revealed that CD4^+^ MAIT cells also produce more IL-2 cytokine in comparison to other cytokine subsets, indicating their potential role in early infection and latency [37]. The CD8^+^ MAIT cell responses were elevated in IGRA^+^ individuals upon early infection with Mtb [38]. Tuberculous pleural effusions are characterized by enhanced MAIT cells expressing IFNγ and TNFα compared to periphery and after stimulation with Bacillus Calmette-Guerin (BCG) [23,39]. Our data suggest that upon stimulation with mycobacterial antigens (PPD and WCL), the geometric means (GMs) of Th1 (IFNγ and TNFα) cytokines expressing MAIT cells were elevated and that of IL-2 cytokine decreased in the LTB-DM and PDM-coinfected group compared to the LTB-NDM group. Moreover, these responses were comparatively pathogen-specific or Mtb-antigen-specific because the increased or decreased regulation of Th1 cytokines among the study individuals was almost totally abrogated in the unstimulated and positive control antigen-stimulation treatments utilized in our experiments. Therefore, herein, we have described the important association of altered MAIT cells expressing Th1 cytokines in LTB-DM and PDM co-morbidity.

In addition to CD4^+^ Th1-expressing cytokines, Th17 cytokines are an important marker in achieving cellular immunity to TB disease [34], and their cellular and molecular mechanisms in the differential production by MAIT cells in LTB-NDM, PDM, and DM comorbidity have been less-explored. Th17 cells produce IL-17A, IL- 17F, and IL-22 cytokines, and play a key role in the defense against TB infection, and are key for immune-mediated protection against hyper-virulent TB strains [23]. Active TB individuals are associated with reduced IFNγ^+^IL-17^+^ dual cytokine production compared to those with LTB disease [36]. Elevated levels of soluble IL-17 cytokine were associated with LTB compared to active TB [40]. IL-17 production is implicated as a distinctive characteristic of MAIT cells [41]. CD8^+^ MAIT cells produce higher IL-17 in IGRA converters than nonconverters, indicating a potential role of IL-17 in LTB [36]. Therefore, restricting IL-17 immune responses might induce protection. We showed that the GMs of MAIT cells expressing IL-17A, IL-17F, and IL-22 cytokines were significantly elevated in PDM- compared to DM-affected LTB individuals after stimulation with WCL and/or PPD mycobacterial antigen. This increase was antigen-specific, and MAIT cells expressing IL-17 family cytokines were significantly different between those with LTB-PDM and/or DM comorbidities. Recently, we also showed that the levels of IFNγ, TNFα, and IL-17A cytokines were significantly higher in the culture supernatants of LTB (DM and PDM) coinfected patients in comparison to LTB-NDM patients [42]. Additionally, with Mtb antigen stimulation, PPD responded the most; one possible reason for this finding might be that WCL and P/I antigen are more highly specific to the CD3^+^ T-cell-specific immune responses than the MAIT (CD161+) cell responses compared to PPD antigen, which more broadly activates diverse immune responses.

Apart from the potential capability of MAIT cells to produce cytokines, they have the ability to induce a cytotoxic effect on Mtb-infected cells, and therefore mediate a crucial function through their antimicrobial effect [43]. MAIT cells spontaneously increase their cytolytic effector role to enhance the proficiency of the elimination of infected cells via the granzyme (GZE)-B- and perforin (PFN)-dependent routes or via granulysin [44,45]. Our investigation showed that MAIT cells expressing cytotoxic (PFN and GZE B) markers and GNLSN (only in PDM, which might be due to GNLSN activation taking more time and, in our study, we stimulated only for a shorter period; therefore, we did not observe major differences in that cytotoxic marker [16]) were significantly reduced upon stimulation with Mtb (PPD and WCL) antigens compared to UNS and P/I antigen stimulation. Specifically, the cytotoxic ability of this marker was greatly compromised in DM and PDM individuals coinfected with LTB disease, suggesting that DM comorbidities might fail to offer the necessary immune protection against TB infection. The UMAP analysis also provided additional evidence and demonstrated the significant expression of the PFN immune cluster, suggesting that DM comorbidity highly influences those markers in LTB individuals. 

Overall, our data demonstrated the role of MAIT cells in LTB and NDM, PDM, and DM comorbidities. Our data clearly displayed that both DM and PDM comorbidities have modified Th1 cytokines and decreased cytotoxic marker immune responses during LTB infection, which can possibly be attributed to higher risk in the disease pathogenesis of LTB infection. However, our study has certain limitations: mainly the small sample size, being cross-sectional and not longitudinal, and the lack of inclusion of healthy controls as an additional group for comparison. Additionally, we did not use live/dead flow antibodies to distinguish dead (early apoptotic cells) from live cells, which is another study limitation. In the future, we aim to study the MAIT cells expressing Th1/Th17 cytokines and cytotoxic markers after blocking with either MR1 tetramer to assess the MR1 dependence activation or stimulation with IL-12/IL-18, IL-15, and IFNα cytokines to address the bystander activation of the above markers. Additionally, our other future interest is performing a proliferation assay upon sorting the MAIT cells of LTB coinfected individuals.

## Figures and Tables

**Figure 1 pathogens-11-00087-f001:**
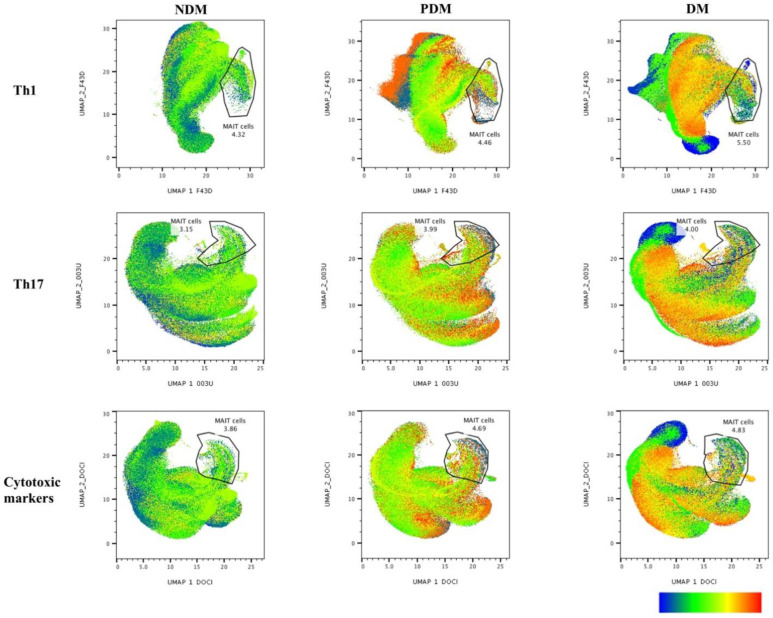
Multi-dimensionality profile showing the percentage of MAIT cells on UMAP analysis. We down sampled (5000 cells) from gated CD3^+^ T cells and performed unsupervised UMAP (UMAP_1 vs. UMAP_2) clusters for latent tuberculosis (LTB) non-diabetes mellitus (NDM), pre-diabetes mellitus (PDM), and diabetes mellitus (DM) individuals, and gated on positive expression of MAIT (TRAV1-2 V⍺7.2 and CD161) cells.

**Figure 2 pathogens-11-00087-f002:**
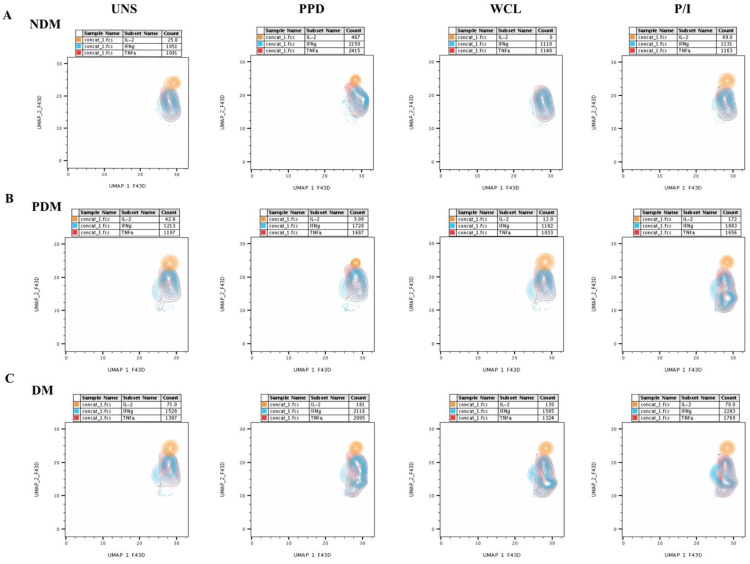
UMAP data on Th1 cytokines expressing MAIT cells. UMAP analysis showing the expression of Th1 cytokines, TNFα (red), IFNγ (aqua), and IL-2 (orange) clusters upon no-antigen UNS, PPD, WCL, and P/I antigen stimulation for the (**A**) LTB-NDM, (**B**) LTB-PDM, and (**C**) LTB-DM groups.

**Figure 3 pathogens-11-00087-f003:**
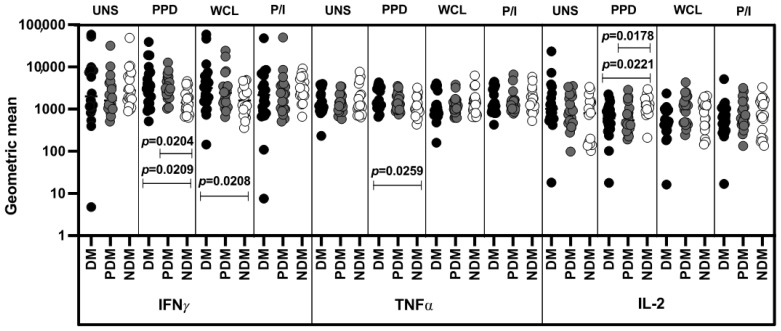
Altered GM of MAIT cells expressing Th1 (IFNγ, IL-2, TNFα) cytokines in LTB comor-bidities. Peripheral blood mononuclear cells (PBMCs) were cultured with media alone or myco-bacterial or positive control antigens for 18 h, and the UNS (geometric mean (GM)) and anti-gen-stimulated GM of Th1 cytokines were determined. The unstimulated, PPD, WCL, and P/I an-tigen-stimulated conditions in LTB-DM (black colored circle, n = 20), LTB-PDM (grey coloured circle, n = 20), and LTB-NDM (white colored circle, n = 20) individuals are displayed. Each circle represents a single individual and the bars represent the geometric mean values. *p*-values were calculated using the Kruskal–Wallis test with multiple Dunn’s comparison.

**Figure 4 pathogens-11-00087-f004:**
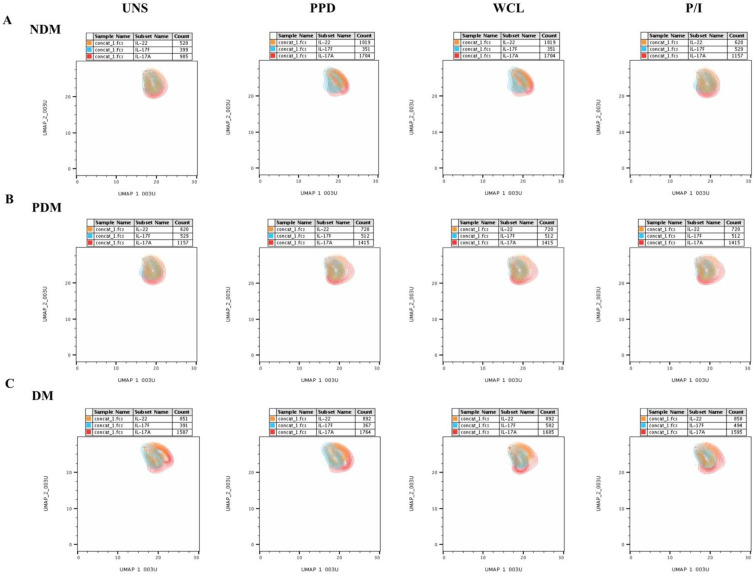
No significant differences in MAIT cells expressing Th17 (IL-17A, IL-17F, IL-22) cytokines on UMAP. UMAP analysis showing the expressions of Th17 cytokine immune IL-17A (red), IL-17F (aqua), IL-22 (orange) clusters upon no-antigen UNS, PPD, WCL, and P/I antigen stimulation among (**A**) LTB-NDM, (**B**) LTB-PDM, and (**C**) LTB-DM groups.

**Figure 5 pathogens-11-00087-f005:**
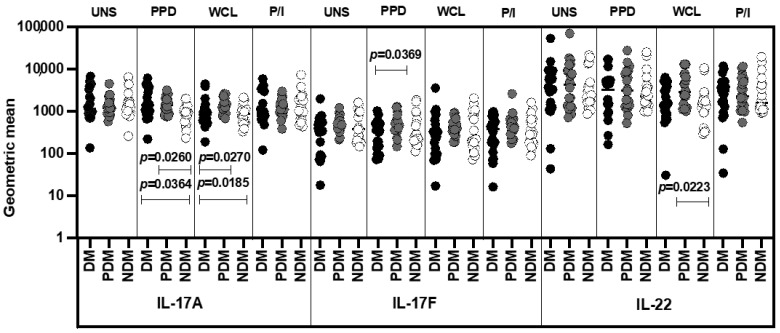
Elevated GM of MAIT cells expressing IL-17 family cytokines in LTB-PDM individuals. Peripheral blood mononuclear cells (PBMCs) were cultured with media alone or mycobacterial or positive control antigens for 18 h, and the UNS and antigen-stimulated GM of Th17 cytokines were determined. The unstimulated, PPD, WCL, and P/I antigen-stimulated conditions in LTB-DM (black colored circle, n = 20), LTB-PDM (grey colored circle, n = 20), and LTB-NDM (white coloured circle, n = 20) individuals are displayed. Each circle represents a single individual and the bars represent the geometric mean values. *p*-values were calculated using the Kruskal–Wallis test with multiple Dunn’s comparisons.

**Figure 6 pathogens-11-00087-f006:**
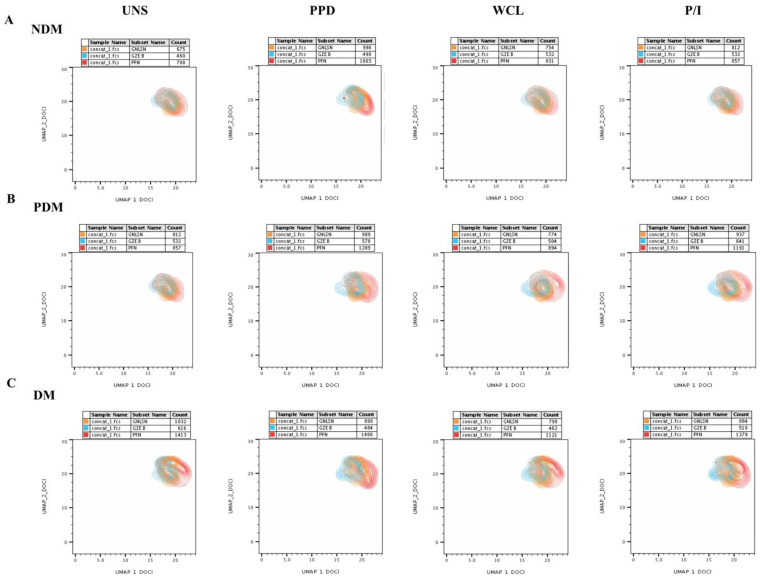
UMAP data on MAIT cells expressing cytotoxic (perforin (PFN), granzyme (GZE) B, and granulysin (GNLSN)) markers. UMAP analysis showing the expression of cytotoxic markers immune PFN (red), GZE B (aqua), and GNLSN (orange) clusters upon no-antigen UNS, PPD, WCL, and P/I antigen stimulation among the (**A**) LTB-NDM, (**B**) LTB-PDM, and (**C**) LTB-DM groups.

**Figure 7 pathogens-11-00087-f007:**
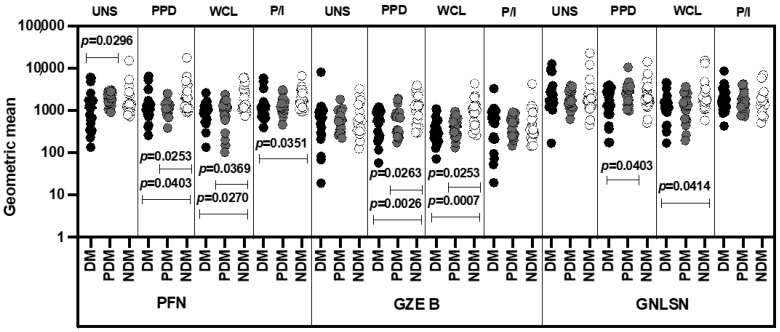
Decreased GM of MAIT cells expressing cytotoxic markers in LTB-DM and/or PDM individuals. Peripheral blood mononuclear cells (PBMCs) were cultured with media alone or mycobacterial or positive control antigens for 18 h, and the UNS and antigen-stimulated cytotoxic markers were determined. The unstimulated, PPD, WCL, and P/I antigen-stimulated conditions in LTB-DM (black colored circle, n = 20), LTB-PDM (grey coloured circle, n = 20), and LTB-NDM (white colored circle, n = 20) individuals are displayed. Each circle represents a single individual and the bars represent the geometric mean values. *p*-values were calculated using the Kruskal–Wallis test with multiple Dunn’s comparisons.

**Figure 8 pathogens-11-00087-f008:**
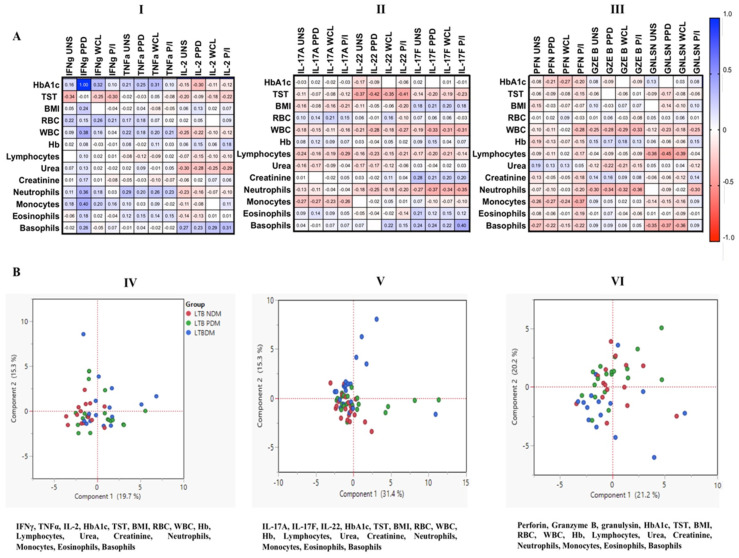
(**A**) [I–III] Spearman correlation analysis and (**B**) [IV–VI] PCA of MAIT cells expressing Th1/Th17 cytokines and cytotoxic markers.

**Table 1 pathogens-11-00087-t001:** Demographics of the study population.

Study Demographics	LTB-NDM	LTB-PDM	LTB DM
Number of subjects recruited (n)	20	20	20
Sex (M/F)	10/10	11/9	11/9
Median age in years (range)	39.6 (24–62)	45.9 (25–62)	47.1 (25–60)
Glycated hemoglobin level, %	<5.48 (5.0–5.7)	>6.1 (5.7–6.3)	8.43 (6.50–11.96)
QuantiFERON-TB gold assay	Positive	Positive	Positive

## Data Availability

All the data are available within the manuscript.

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
