# Peer review of "Multi-Dimensionality Immunophenotyping Analyses of MAIT Cells Expressing Th1/Th17 Cytokines and Cytotoxic Markers in Latent Tuberculosis Diabetes Comorbidity"

_pathogens, 2022, doi:10.3390/pathogens11010087_

Round 1

Reviewer 1 Report

the work of  Raj Kathamuthu e collegues  shows different Th1 and cytotoxic marker population clusters and increased Th1, IL-17A cytokines, and diminished cytotoxic markers expressing MAIT cell frequencies are associated with Latent TB PDM and DM comorbidities. Interesting and well-structured results, but some deepening would be necessary. especially to corroborate the statistical significance.
It would be necessary to carry out the analysis of the principal components (PCA analysis), to highlight among the different groups which markers are more important. Moreover, the analysis of UMAP, although nice to see, does not highlight the differences, a table with the values and significances is necessary to add.

Have there been any correlations between the expression of the markers and the clinical data of the patients? it would be very interesting to highlight how the immune response is correlated with symptoms, severity, biochemical markers in the different groups.

Finally, a graphical abstract highlighting the differences in expression in the groups and also any correlations with the clinic is suggested

minor point: Explain more in materials and methods the number of lymphocytes analyzed, were stimulations done in 96-well plates?

Also, there is a repetition in lines 132/133 and other editing errors throughout the text

Author Response

The work of  Raj Kathamuthu e collegues  shows different Th1 and cytotoxic marker population clusters and increased Th1, IL-17A cytokines, and diminished cytotoxic markers expressing MAIT cell frequencies are associated with Latent TB PDM and DM comorbidities. Interesting and well-structured results, but some deepening would be necessary. especially to corroborate the statistical significance.

It would be necessary to carry out the analysis of the principal components (PCA analysis), to highlight among the different groups which markers are more important.

Response: As suggested by the reviewer, now we have included the PCA analysis in the manuscript as Figure 8B [IV-VI].

Moreover, the analysis of UMAP, although nice to see, does not highlight the differences, a table with the values and significances is necessary to add.

Response: As suggested by the reviewer, now we have included UMAP statistical values in supplementary table 1.

Have there been any correlations between the expression of the markers and the clinical data of the patients? it would be very interesting to highlight how the immune response is correlated with symptoms, severity, biochemical markers in the different groups.

Response: As suggested by the reviewer, now we have included the correlation analysis as Figure 8A [I-III] (P values given in Supplementary Table 2).

Finally, a graphical abstract highlighting the differences in expression in the groups and also any correlations with the clinic is suggested

Response: As suggested by the reviewer, now we have included the graphical abstract.

minor point: Explain more in materials and methods the number of lymphocytes analyzed, were stimulations done in 96-well plates?

Response: We have acquired 50,000 gated lymphocyte events and performed the analysis. The stimulation was performed using 12 well culture plates. It is now included in the materials and methods section.

Also, there is a repetition in lines 132/133 and other editing errors throughout the text

Response: We have now revised the errors throughout the text.

Reviewer 2 Report

The authors have submitted a manuscript dealing with the detection of MAIT cells in peripheral blood and their functions. The background is a possible involvement of these cells in tuberculosis and diabetes mellitus, two diseases that may be associated with each other. In the introduction, these connections are clearly deduced. 
In the M&M chapter, the patients and the study design are presented. The number is appropriate. Peripheral blood cells are isolated and frozen. Note 1: Has it been validated that the freeze-thaw process does not change the properties of the cells studied? Cells were cultured, stimulated and analyzed by flow cytometry. Data analysis was performed using Flowjo 3 with the appropriate plugins. The statistics are briefly described. 
In the Results chapter, the flow cytometric data are shown with several figures. Note 2: Unfortunately, the arrangement of the graphs in Figure 2 is quite awkward. Note 3: Can the cytokine expressions in Figure 3 be verified in the preserved culture supernatant? Note 4: The differences in the statements in Figures 4 and 5 (no differences vs. single significant differences) is confusing. Cytotoxicity data follow. The results are briefly interpreted in the Discussion.  

Author Response

The authors have submitted a manuscript dealing with the detection of MAIT cells in peripheral blood and their functions. The background is a possible involvement of these cells in tuberculosis and diabetes mellitus, two diseases that may be associated with each other. In the introduction, these connections are clearly deduced.

In the M&M chapter, the patients and the study design are presented. The number is appropriate. Peripheral blood cells are isolated and frozen.

Note 1: Has it been validated that the freeze-thaw process does not change the properties of the cells studied?

Response: Yes, after thawing the cell were rested for 2 hours and then the cell viability was assessed using the trypan blue exclusion assay. It is now included in the materials and methods section.

Cells were cultured, stimulated and analyzed by flow cytometry. Data analysis was performed using Flowjo 3 with the appropriate plugins. The statistics are briefly described.

In the Results chapter, the flow cytometric data are shown with several figures.

Note 2: Unfortunately, the arrangement of the graphs in Figure 2 is quite awkward.

Response: We apologize and now we have revised the figures (we have now removed the histogram since it is another way of representation to that of the main figure) and shown in Figure 2.

Note 3: Can the cytokine expressions in Figure 3 be verified in the preserved culture supernatant?

Response: We previously shown the data for TNFa, IFNg and IL-17A using the preserved culture supernatants. This is now included in the discussion section.

Reference number 44. Gokul Raj et al., Decreased Frequencies of Gamma/Delta T Cells Expressing Th1/Th17 Cytokine, Cytotoxic, and Immune Markers in Latent Tuberculosis-Diabetes/Pre- Diabetes Comorbidity. Front. Cell. Infect. Microbiol. 2021; 11:756854.

Note 4: The differences in the statements in Figures 4 and 5 (no differences vs. single significant differences) is confusing. Cytotoxicity data follow. The results are briefly interpreted in the Discussion. 

Response: Now we have corrected the sentence, in Figure 4, the data were not significant upon high dimensionality analysis. In Figure 5, the frequencies of IL-17 and IL-22 cytokines expressing MAIT cells were significantly different.

Reviewer 3 Report

In this paper the authors investigate cytokine and cytotoxic-markers expressing MAIT cells in latent tuberculosis diabetes comorbitity.

The authors provide a relevant study examining this important cell-type and its reactivity via flow cytometric analysis of PBMCs from subjects with latent tuberculosis, which are either normoglycemic or have pre-diabetes or diabetes mellitus.

The flow-cytometric approach of the authors aims to answer the research questions, yet falls short on data- presentation and analysis. Major revision is therefore required to improve its understanding and show the validity of data.

  1. The authors analyse the cytokine-expressing cells with UMAP projections in low quality and small figures. The full better quality figures are added extra, yet should be in the paper. What advantage does an UMAP hold over histograms, which are presented next to it? Additionally, net frequencies are hard to understand. The authors are advised to present their data either with MFI (or geo mean) or alternatively as % of positive cells, as they nicely show in supplemental data. Higher quality of figures or a simpler way of presenting (histogram/overlay) is a must for the reader’s understanding.

2.) How do the authors explain the relatively low numbers of cells reacting to stimulation e.g. to WCL or P/I?

3.) Does the % of cytokine positive cells have a functional effect? The authors should confirm their findings with for example a cytokine release assay. MAIT cells can be sorted or selected (Miltenyi beads) and used further in for example proliferation assays. A functional result in different reactivity of Pre-DM or DM individuals would add credit to their FC data.

3.) As it is known that MAIT are not high in frequency, why did the authors acquire ONLY 50 000 cells in the lymphocyte gate?

4.) The authors state that PBMC's taken from the study population have been frozen and thawed, before analysis. They claim dead cells were assessed by Trypan blue. However, this does not exclude them in further analysis and they do not report viability for each sample. Moreover, they do not include a viability dye in their flow-cytometric analysis. This would give a more correct percentage of positive cells following stimulation. Maybe low numbers in some gates were due to non-responsive/dead cells.

Minor: The immunophenotyping analysis the authors demonstrate is not high/multi dimension. In order to be called as such, the authors would need to use 10+ markers.

Author Response

The authors analyse the cytokine-expressing cells with UMAP projections in low quality and small figures. The full better-quality figures are added extra, yet should be in the paper. What advantage does an UMAP hold over histograms, which are presented next to it? Additionally, net frequencies are hard to understand. The authors are advised to present their data either with MFI (or geo mean) or alternatively as % of positive cells, as they nicely show in supplemental data. Higher quality of figures or a simpler way of presenting (histogram/overlay) is a must for the reader’s understanding.

Response: As suggested by the reviewer, histograms did not provide any additional information than the UMAP figure, thus we have now removed the histogram and shown the figure using UMAP figure alone. Also we have included the figures using the geo mean values instead of absolute frequencies.

2.) How do the authors explain the relatively low numbers of cells reacting to stimulation e.g. to WCL or P/I?.

Response: We apologise and did not know the exact reason; however, one possible reason might be WCL and P/I antigen were highly specific to the CD3+ T cell specific immune responses than the MAIT (CD161+) cell responses compared to PPD antigen which has broader activation of diverse immune responses.

3.) Does the % of cytokine positive cells have a functional effect? The authors should confirm their findings with for example a cytokine release assay. MAIT cells can be sorted or selected (Miltenyi beads) and used further in for example proliferation assays. A functional result in different reactivity of Pre-DM or DM individuals would add credit to their FC data.

Response: We apologise and we did not have the PBMC samples to perform the sorting experiment. We included this as a future interest in the discussion section as “Additionally, the other interest to perform the proliferation assay upon sorting the MAIT cells of LTB coinfected individuals”.

3.) As it is known that MAIT are not high in frequency, why did the authors acquire ONLY 50 000 cells in the lymphocyte gate?

Response: Some of our PBMCs has less number of cells and to maintain the similarity in performing the sample analysis in flow jo we have acquired only 50,000 cells.

4.) The authors state that PBMC's taken from the study population have been frozen and thawed, before analysis. They claim dead cells were assessed by Trypan blue. However, this does not exclude them in further analysis and they do not report viability for each sample. Moreover, they do not include a viability dye in their flow-cytometric analysis. This would give a more correct percentage of positive cells following stimulation. Maybe low numbers in some gates were due to non-responsive/dead cells.

Response: We agree with the reviewer and gated the single cells through FSC A versus FSC H gating prior to gate the positive MAIT cells. It is now included in the materials and methods section.

Minor: The immunophenotyping analysis the authors demonstrate is not high/multi dimension. In order to be called as such, the authors would need to use 10+ markers.

Response: We agree with the reviewer and based on the reviewer comments now we have revised our title as “Dimensionality immunophenotyping analyses of MAIT cells expressing Th1/Th17 cytokines and cytotoxic markers in latent tuberculosis diabetes comorbidity”.

Round 2

Reviewer 1 Report

The authors have been respond to all my suggestions.

Author Response

NIL

Reviewer 2 Report

Thank you for consifering reviewers' comments.

Author Response

NIL

Reviewer 3 Report

The authors have sufficiently addressed some of the concerns. The study is also further improved with the added correlations of cytokine-expressing cells and some patient-related parameters such as HbA1c.  However, some questions pertinent to the study quality remain unanswered.

2.) How do the authors explain the relatively low numbers of cells reacting to stimulation e.g. to WCL or P/I?.

Response: We apologise and did not know the exact reason; however, one possible reason might be WCL and P/I antigen were highly specific to the CD3+ T cell specific immune responses than the MAIT (CD161+) cell responses compared to PPD antigen which has broader activation of diverse immune responses.

Further:  Can the authors prove that CD3+ lymphocytes in respond to P/I in their setup? They should add this information (% of CD3+ cells responding to P/I) as a supplemetary figure to confirm, that P/I IS a positive control. 

4.) The authors state that PBMC's taken from the study population have been frozen and thawed, before analysis. They claim dead cells were assessed by Trypan blue. However, this does not exclude them in further analysis and they do not report viability for each sample. Moreover, they do not include a viability dye in their flow-cytometric analysis. This would give a more correct percentage of positive cells following stimulation. Maybe low numbers in some gates were due to non-responsive/dead cells.

Response: We agree with the reviewer and gated the single cells through FSC A versus FSC H gating prior to gate the positive MAIT cells. It is now included in the materials and methods section.

Further:  Including only 'single' cells is the correct analysis step, however this does not provide correct information regarding viability of cells, which is directly connected to the responsiveness of cells to stimulation. Authors SHOULD list assessed viability range by tryptan blue assay following thawing of PBMCs (80?, 90? >99%?) in the materials and methods section. Additionally authors should list this analysis step as a major limitation step in their discussion, since correct flow cytometric analysis in stimulation and/or activation assays includes removing dead or early apoptotic cells from further analysis.

Minor:

  • English spell still recquired. For example in line: 217
  • Significance chosen for Spearman correlation should be listed in Figure 8 legend.

Author Response

The authors have sufficiently addressed some of the concerns. The study is also further improved with the added correlations of cytokine-expressing cells and some patient-related parameters such as HbA1c. However, some questions pertinent to the study quality remain unanswered.

2.) How do the authors explain the relatively low numbers of cells reacting to stimulation e.g. to WCL or P/I?.

Response: We apologise and did not know the exact reason; however, one possible reason might be WCL and P/I antigen were highly specific to the CD3+ T cell specific immune responses than the MAIT (CD161+) cell responses compared to PPD antigen which has broader activation of diverse immune responses.

Further:  Can the authors prove that CD3+ lymphocytes in respond to P/I in their setup? They should add this information (% of CD3+ cells responding to P/I) as a supplementary figure to confirm, that P/I IS a positive control.

Response: As suggested by the reviewer, “The percentage of CD3+ population between UNS and P/I antigen stimulation were shown in Supplementary Materials Figure S3” in the second paragraph of the results section.

4.) The authors state that PBMC's taken from the study population have been frozen and thawed, before analysis. They claim dead cells were assessed by Trypan blue. However, this does not exclude them in further analysis and they do not report viability for each sample. Moreover, they do not include a viability dye in their flow-cytometric analysis. This would give a more correct percentage of positive cells following stimulation. Maybe low numbers in some gates were due to non-responsive/dead cells.

Response: We agree with the reviewer and gated the single cells through FSC A versus FSC H gating prior to gate the positive MAIT cells. It is now included in the materials and methods section.

Further:  Including only 'single' cells is the correct analysis step, however this does not provide correct information regarding viability of cells, which is directly connected to the responsiveness of cells to stimulation. Authors SHOULD list assessed viability range by tryptan blue assay following thawing of PBMCs (80?, 90? >99%?) in the materials and methods section. Additionally, authors should list this analysis step as a major limitation step in their discussion, since correct flow cytometric analysis in stimulation and/or activation assays includes removing dead or early apoptotic cells from further analysis.

Reply: Overall, we observed 80-85% cell confluency in our study cohort. This is now included in the materials and methods section.

Also, we did not use live/dead flow antibodies to distinguish dead (early apoptotic cells) from live cells is the other study limitation. This is now included in the discussion section.

Minor:

English spell still required. For example, in line: 217

Response: As suggested by the reviewer, we have now revised the corrected the english throughout the text.

 Significance chosen for Spearman correlation should be listed in Figure 8 legend.

Response: P<0.05 considered as the significant. This is now included in the figure legend 8 and in the materials and methods.